# Oxidative insult can induce malaria-protective trait of sickle and fetal erythrocytes

Marek Cyrklaff[1], Sirikamol Srismith[1], Britta Nyboer[1], Kvetoslava Burda[2], Angelika Hoffmann[3,4], Felix Lasitschka[5,6], Sophie Adjalley[7], Cyrille Bisseye[8], Jacques Simpore[8], Ann-Kristin Mueller[1,5], Cecilia P. Sanchez[1], Friedrich Frischknecht[1] & Michael Lanzer[1,5]

*Plasmodium falciparum* infections can cause severe malaria, but not every infected person develops life-threatening complications. In particular, carriers of the structural haemoglobinopathies S and C and infants are protected from severe disease. Protection is associated with impaired parasite-induced host actin reorganization, required for vesicular trafficking of parasite-encoded adhesins, and reduced cytoadherence of parasitized erythrocytes in the microvasculature. Here we show that aberrant host actin remodelling and the ensuing reduced cytoadherence result from a redox imbalance inherent to haemoglobinopathic and fetal erythrocytes. We further show that a transient oxidative insult to wild-type erythrocytes before infection with *P. falciparum* induces the phenotypic features associated with the protective trait of haemoglobinopathic and fetal erythrocytes. Moreover, pretreatment of mice with the pro-oxidative nutritional supplement menadione mitigate the development of experimental cerebral malaria. Our results identify redox imbalance as a causative principle of protection from severe malaria, which might inspire host-directed intervention strategies.

[1] Center of Infectious Diseases, Parasitology, Heidelberg University Hospital, Im Neuenheimer Feld 324, 69120 Heidelberg, Germany. [2] Faculty of Physics and Applied Computer Science, AGH—University of Science and Technology in Kraków, Al. Mickiewicza 30, 30-059 Kraków, Poland. [3] Department of Neuroradiology, Heidelberg University Hospital, Im Neuenheimer Feld 400, 69120 Heidelberg, Germany. [4] Division of Experimental Radiology, Department of Neuroradiology, Heidelberg University Hospital, Im Neuenheimer Feld 400, 69120 Heidelberg, Germany. [5] German Centre for Infection Research (DZIF), Partner Site Heidelberg, 69120 Heidelberg, Germany. [6] Institute of Pathology, Heidelberg University Hospital, Im Neuenheimer Feld 224, 69120 Heidelberg, Germany. [7] European Molecular Biology Laboratory (EMBL), Genome Biology Unit, Meyerhofstrasse 1, 69117 Heidelberg, Germany. [8] Biomolecular Research Center Pietro Annigoni, University of Ouagadougou, 01 BP 364 Ouagadougou, Burkina Faso. Correspondence and requests for materials should be addressed to M.C. (email: marek.cyrklaff@med.uni-heidelberg.de) or to M.L. (email: michael.lanzer@med.uni-heidelberg.de).

Sickle cell trait is a common hereditary blood disorder resulting from a single amino-acid substitution of glutamic acid to valine at position 6 in the β-globin subunit ($\beta^{6glu \to val}$) of haemoglobin. Sickle cell trait can cause severe microvascular dysfunction when inherited from both parents, but it affords a survival benefit to the heterozygous carrier in infections with the human malaria parasite *Plasmodium falciparum*[1]. Similarly, carriers of haemoglobin C (HbC; $\beta^{6glu \to lys}$) and newborns within the first months of life until haemoglobin F production ceases are protected from severe malaria[2–4]. The mechanism by which haemoglobin S (HbS), HbC, and fetal haemoglobin (HbF) bestow a selective advantage in *P. falciparum* infections is still a matter of debate[5–7]. However, recent evidence has suggested that protection from, and pathology of, malaria are closely linked together through a common mechanistic network.

The pathology of tropical malaria is associated with the cytoadhesive properties of *P. falciparum*-infected erythrocytes that sequester in the deep vascular bed of inner organs, such as the brain, to escape splenic clearance. The cytoadhering erythrocytes interfere with vital functions of the circulatory system and, in addition, can elicit life-threatening inflammatory reactions in the microvasculature[8]. These pathological sequelae seem mitigated in carriers of the structural haemoglobinopathies S and C and in newborns as parasitized erythrocytes containing these haemoglobin variants display a reduced capacity to cytoadhere to human microvascular endothelial cells[2,3,9–12].

Recent findings have suggested that reduced cytoadherence results from impaired intracellular trafficking and sorting of parasite-encoded adhesins to the host cell surface[13,14]. According to our current understanding, adhesins are usually trafficked across the host cell cytoplasm to the erythrocyte surface via a complex pathway that involves transient association with parasite-generated unilamellar membrane profiles, termed Maurer's clefts, and vesicular transport along actin-containing filaments from the Maurer's clefts to knob-like protrusions in the host cell membrane, where the adhesins are presented[10,15,16]. In addition to vesicles budding off from the Maurer's clefts, there is evidence of transport vesicles originating from the parasitophorous vacuolar membrane that separates the parasite from the host erythrocyte[10,17,18].

The actin-containing filaments are of host origin and are generated by the parasite by reorganizing the actin of the erythrocyte's membrane skeleton[10]. Parasite-induced actin remodelling is impaired in parasitized homozygous HbCC and heterozygous HbSC erythrocytes, resulting in untypical short actin filaments that do not connect the Maurer's clefts with the knobs[10]. Moreover, Maurer's clefts are amorphous and they display an unusual stage-dependent motion pattern[10,19]. In addition, knobs are enlarged and dispersed[3,9,10,20].

While impaired host actin remodelling has been demonstrated for parasitized HbCC and HbSC erythrocytes, it is unclear whether this aberration extends to include erythrocytes from newborns and from heterozygous HbAS and HbAC carriers. The HbAS and HbAC alleles are approximately six times more prevalent in malaria-endemic regions than the rarer HbCC and HbSC genotypes[21]. The respective carriers of HbAS and HbAC benefit the most from the presence of these polymorphisms since they are protected from the life-threatening complications of malaria, while having no or only mild symptoms. Here we investigate the molecular mechanism(s) underpinning impaired host actin remodelling and reduced cytoadhesion. We show that aberrant host actin remodelling and reduced cytoadherence are steps in a cascade of events that starts with the redox imbalance inherent to haemoglobinopathic and fetal erythrocytes. In support of this conclusion we show that transiently exposing wild-type erythrocytes to a pro-oxidative environment before infection with *P. falciparum* induces the phenotypic features associated with the protective trait of haemoglobinopathic and fetal erythrocytes. Our results provide novel insights into a naturally occurring protective mechanism against severe malaria and identify redox imbalance as a causative principle of protection. We anticipate that our findings will inspire intervention strategies that target the host cell rather than the parasite itself, and, thus, might be exempt from parasite-encoded drug resistance mechanisms.

## Results

**Impaired host actin remodelling**. The *P. falciparum* strain FCR3$^{CSA}$ established lamellar Maurer's clefts, small knobs and an elaborate actin network in wild-type HbAA erythrocytes (Fig. 1a), consistent with previous reports using the *P. falciparum* strains HB3 and CS2 (refs 10,22). In comparison, cryo-tomographic images of HbAS and HbAC erythrocytes infected with the same FCR3$^{CSA}$ strain revealed malformed Maurer's clefts, enlarged knobs and short, unattached actin filaments (Fig. 1b,c). Comparable deviations were observed in parasitized erythrocytes from fetal blood (Fig. 1d). The aberrations observed were comparable to those seen in parasitized HbSC and HbCC erythrocytes[10].

In uninfected HbAA erythrocytes, actin is organized as short protofilaments of 30–40 nm that, together with other factors, form junctional complexes, which crosslink spectrin tetramers into hexagonal arrays[23]. A quantitative analysis of six tomograms from each condition confirmed the differences in actin organization between the different parasitized erythrocyte variants and also between the corresponding infected and uninfected red blood cells (RBCs; Fig. 2a). In all cases, the actin filaments tended to be longer in infected erythrocytes as compared with their uninfected counterparts (which in the case of HbAA, HbAS and HbF erythrocytes reached statistical significance, $P < 0.01$ according to Kruskal–Wallis one-way analysis of variance test; Fig. 2a). However, actin filaments in parasitized HbAS, HbAC and HbF erythrocytes were always consistently shorter than in parasitized HbAA erythrocytes ($P < 0.01$ according to Kruskal–Wallis one-way analysis of variance test; Fig. 2a). When comparing the actin length of uninfected erythrocytes, we noted a statistically significant difference between HbAA on the one hand and HbAS, HbAC and HbF RBCs on the other hand ($P < 0.01$ according to Kruskal–Wallis one-way analysis of variance test; Fig. 2a), consistent with a dysfunctional cytoskeleton in these cells, as has previously been demonstrated for uninfected HbAS erythrocytes[24].

Scanning and transmission electron microscopic images confirmed the amorphous structure of Maurer's clefts and the aberrant appearance of knobs in parasitized HbAS, HbAC and HbF erythrocytes (Supplementary Figs 1 and 2). A semi-quantitative analysis of these data revealed that the atypical vesicular morphology of the Maurer's clefts resulted from a volume expansion at constant surface area (Fig. 2b, Supplementary Fig. 2 and Supplementary Table 1), that is, the Maurer's clefts change their shape from a fattened lamellar profile with a low volume to surface area ratio to a vesicular form, consistent with previous reports[10]. The knobs decorating parasitized haemoglobinopathic and fetal erythrocytes had an approximately threefold larger diameter and their density was approximately fivefold reduced, as compared with parasitized HbAA erythrocytes (Fig. 2c and Supplementary Fig. 1), consistent with previous reports[2,3,9,20].

When grown in HbAA erythrocytes, the *P. falciparum* strain FCR3$^{CSA}$ efficiently bound to surfaces coated with chondroitin-4-sulphate (CSA) (Fig. 2d), a model receptor for cytoadhesion of parasitized erythrocytes in the placental intervillous space[25].

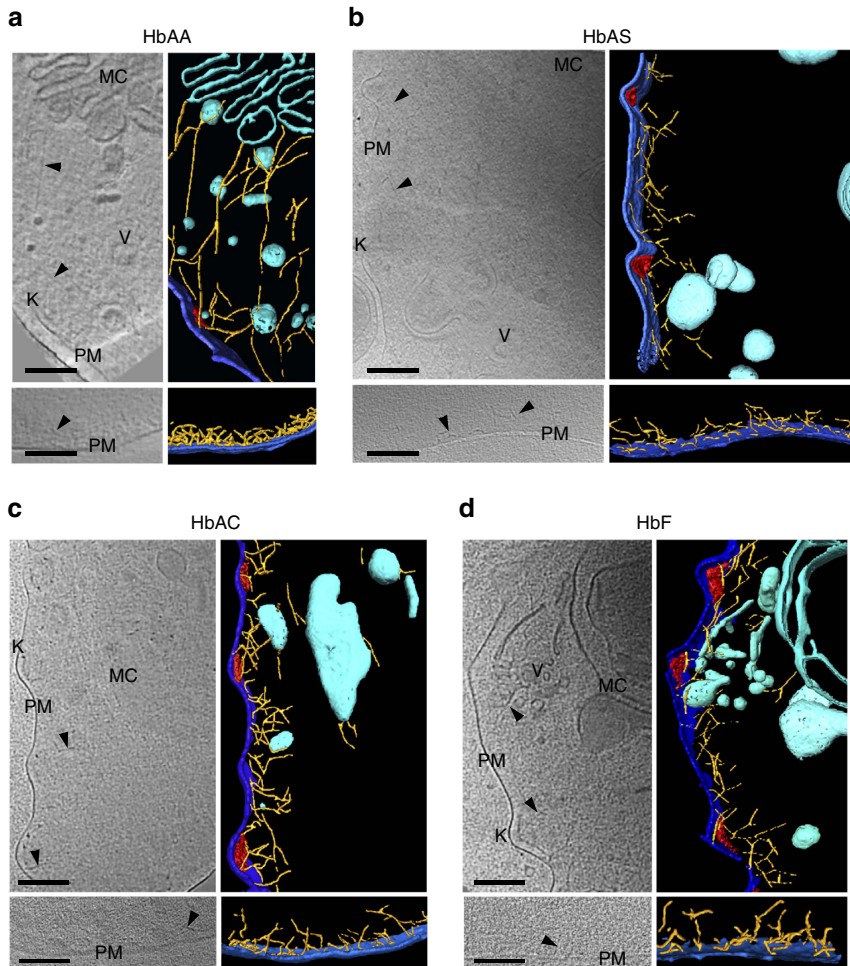

**Figure 1 | Structural features in the cytoplasm of *Plasmodium falciparum*-infected erythrocytes containing different haemoglobin variants.** Cryo-electron tomograms represented by projections of 50 nm-thick slices (left panels) and surface-rendered views of reconstructed volumes (right panels) are shown. Infected erythrocytes at the trophozoite stage (24–32 h post invasion) were investigated throughout the study. *P. falciparum*-infected (top panels) and -uninfected (bottom panels) HbAA (**a**), HbAS (**b**), HbAC (**c**) and HbF erythrocytes (**d**). Representative examples of six tomograms are shown for each case. K, knobs (red); MC, Maurer's clefts (cyan); PM, erythrocyte plasma membrane (dark blue); V, vesicles (cyan); actin filaments (yellow) indicated by arrowheads in the sections. Scale bars, 100 nm.

In comparison, the same parasite strain conferred only a moderate binding capacity to CSA when grown in haemoglobinopathic or fetal erythrocytes (Fig. 2d), as demonstrated in parallel adhesion assays, using FCR3[CSA] parasite populations preselected for CSA binding before infection of the different RBC variants. The different erythrocytes had no significant effect on parasite growth as indicated by comparable multiplication rates and comparable levels of haemozoin, a product of haemoglobin digestion (Fig. 2e). Moreover, the parasite expressed comparable mRNA levels of *var2csa*, the gene encoding the CSA-binding adhesin (Fig. 2f). However, the amount of surface-presented VAR2CSA adhesin was significantly reduced in parasitized haemoglobinopathic and HbF erythrocytes compared with HbAA erythrocytes ($P < 0.01$ according to Kruskal–Wallis one-way analysis of variance test; Fig. 2f), consistent with their impaired adhesion phenotype and other reports using parasite strains expressing adhesin variants other than VAR2CSA[2,3,9].

**Oxidative stress as an underpinning mechanism of protection.** The finding of common phenotypic aberrations in parasitized haemoglobinopathic and fetal erythrocytes points towards a conserved underpinning mechanism. A feature common to these erythrocyte variants is an intrinsic redox imbalance[26,27]. Erythrocytes produce a steady flow of reactive oxygen species under natural physiological conditions, initiated by the spontaneous autoxidation of haemoglobin to methaemoglobin[28]. The superoxide anion radical generated in this process dismutates to hydrogen peroxide, which can subsequently react with haemoglobin to ferryl-haemoglobin or irreversibly oxidized haemichromes. Hydrogen peroxide can further disintegrate methaemoglobin, following oxidation, into a globin-based radical and a ferryl haem group[28]. Sickle cell erythrocytes generate approximately twofold higher quantities of reactive oxygen species due to the instability of HbS[26]. As a consequence, the levels of methaemoglobin and irreversibly oxidized haemichromes are significantly elevated in HbAS erythrocytes relative to wild-type erythrocytes[29], as verified in our study by Mössbauer spectroscopy ($P < 0.01$ according to Kruskal–Wallis one-way analysis of variance test; Table 1). Similarly, elevated levels of methaemoglobin and irreversibly oxidized haemichromes were observed in HbAC and HbF erythrocytes (Table 1)[3].

Oxidative stress can have an impact on many biological processes and is considered the main cause of the pathological consequences associated with sickle cell disease[26]. It is therefore possible that oxidative stress underpins, or at least

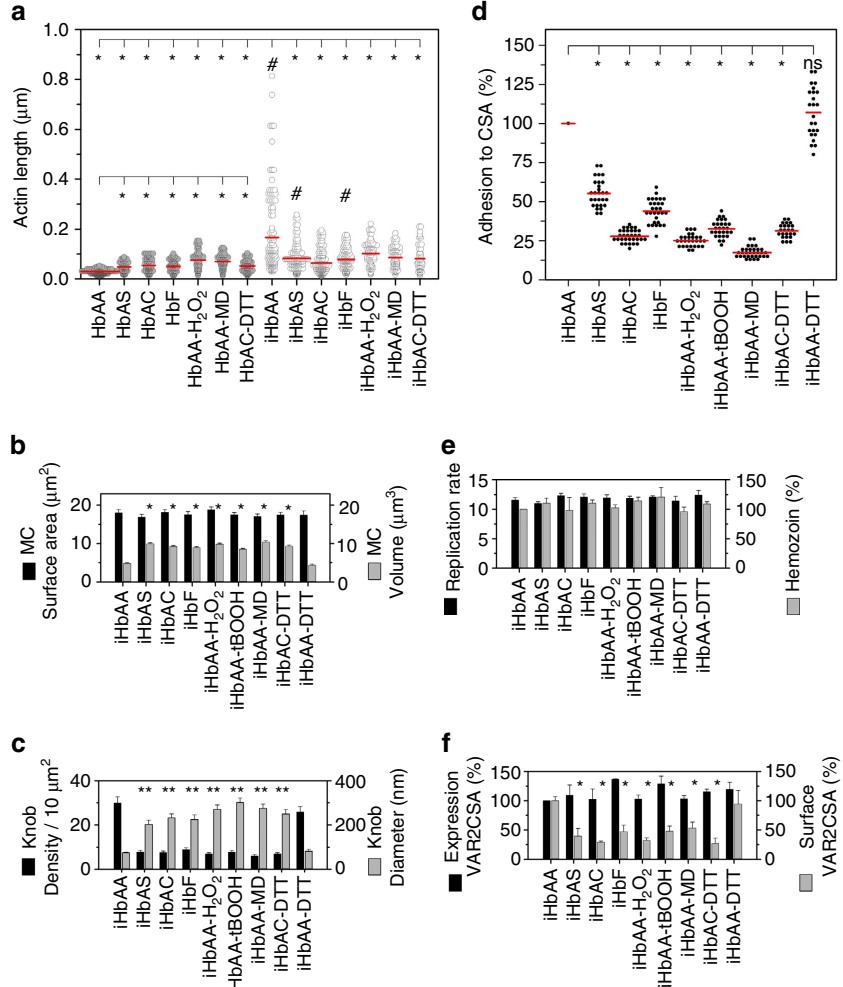

**Figure 2 | Phenotypic features of parasitized erythrocytes associated with natural and induced protection from severe malaria.** The parasitized erythrocyte containing HbAA, HbAS, HbAC and HbF, and erythrocytes pretreated with the redox-altering agents $H_2O_2$, tBOOH, menadione (MD) and DTT before infection with the *P. falciparum* strain FCR3[CSA] were investigated. Note that only the host cells were exposed to these oxidants (before infection) but not the parasites. The prefix 'i' stands for infected red blood cells. (**a**) Frequency histogram of actin filament lengths in uninfected (filled circles) and infected (open circles) erythrocytes. $n > 100$ filaments in each case. Red lines indicate the mean. $*P < 0.001$ compared with uninfected HbAA erythrocytes (lower analysis) or parasitized HbAA erythrocytes (upper analysis), according to Kruskal–Wallis one-way analysis of variance test. #$P < 0.01$ between infected erythrocyte and the uninfected counterpart, according to Kruskal–Wallis one-way analysis of variance test. (**b**) Surface area and estimated volume of Maurer's clefts. Means ± s.e.m. of at least 50 independent determinations are shown for each condition. $*P < 0.001$ compared with infected HbAA erythrocytes, according to Kruskal–Wallis one-way analysis of variance test. NS, not significant. (**c**) Knob size and density as determined from cryo-tomograms and scanning electron microscopy images. $n > 60$ knobs in each case. $*P < 0.001$ compared with infected HbAA erythrocytes, according to Kruskal–Wallis one-way analysis of variance test. (**d**) Frequency histogram showing adherence of parasitized erythrocytes to chondroitin-4-sulfate A, normalized to parasitized HbAA erythrocytes. Each data point represents an independent biological replicate. $*P < 0.001$ compared with infected HbAA erythrocytes, according to Kruskal–Wallis one-way analysis of variance test. (**e**) Parasite replication rates and haemozoin production (normalized to infected HbAA erythrocytes). Means ± s.e.m. of at least 10 independent biological replicates are shown. (**f**) *var2csa* expression levels and surface-presented VAR2CSA adhesin, normalized to infected HbAA erythrocytes. Means ± s.e.m. of at least 5 biological replicates are shown. $*P < 0.001$ compared with infected HbAA erythrocytes, according to Kruskal–Wallis one-way analysis of variance test.

contributes to, the aberrant organization of the protein trafficking pathway and the impaired formation of the cytoadhesion complex, thus providing a mechanism by which haemoglobinopathic and HbF RBCs protect individuals from severe malaria. This hypothesis is all the more attractive since oxidative stress induced by reactive oxygen species (including pro-oxidative products of haemoglobin oxidation, such as ferryl-haemoglobin and globin radicals) can interfere with actin polymerization and actin dynamics[10,30–32]. These effects are explained by conformational changes in the tertiary structure of actin induced by the oxidation of cysteines and the formation of disulphide bonds[33].

To investigate the question of whether oxidative stress is causatively linked with impaired actin remodelling and diminished cytoadherence, we exposed uninfected HbAA erythrocytes to a transient oxidative insult before infection with the *P. falciparum* strain FCR3[CSA]. To this end, we treated uninfected HbAA erythrocytes for 60 min with 1 mM hydrogen peroxide ($H_2O_2$) or 1 mM tert-butyl hydroperoxide (tBOOH)[34,35]. Cells were subsequently washed in buffer containing equimolar concentrations of dithiothreitol (1 mM dithiothreitol (DTT)) to restore reducing conditions and heal those oxidative damages that are reversible[36]. Irreversibly oxidized species remain oxidized under these conditions.

**Table 1 | Levels of oxidized haemoglobins in uninfected erythrocytes.**

| Erythrocyte variant | Pretreatment reagent | Oxy-Hb (%) | Met-Hb (%) | Irreversibly oxidized Hb (%) |
| --- | --- | --- | --- | --- |
| HbAA | — | 94.0 | 1.0 | 5.0 |
| HbAS | — | 85.6 | 2.3 | 12.2 |
| HbAC | — | 83.5 | 3.9 | 12.6 |
| HbF | — | 84.7 | 3.1 | 12.2 |
| HbAA | $H_2O_2$ | 78.1 | 1.3 | 20.6 |
| HbAA | tBOOH | 77.7 | 5.4 | 16.9 |
| HbAA | Menadione | 70.1 | 7.3 | 22.0 |

Met-Hb, methaemoglobin; Oxy-Hb, oxy-haemoglobin.
Where indicated HbAA erythrocytes were pretreated with $H_2O_2$, tert-butyl hydroperoxide (tBOOH) or menadione. The experimental uncertainty of the determinations is <1.5% and the mean of three independent biological replicates, using blood from three different donors, is shown.

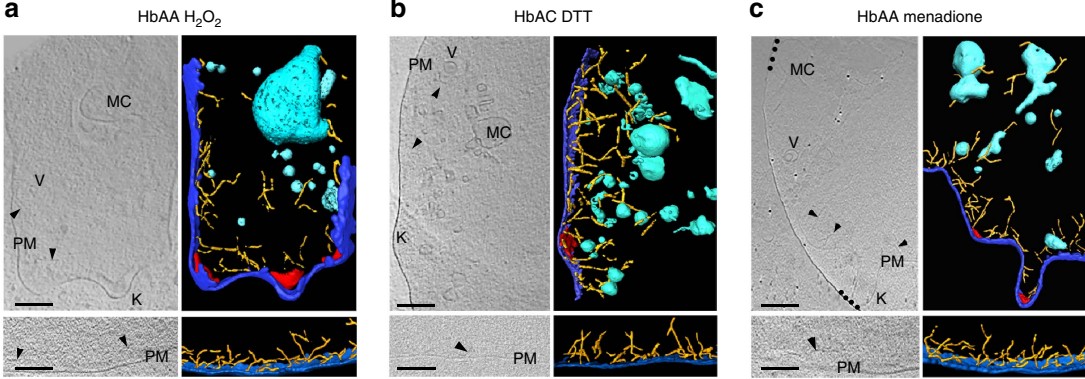

**Figure 3 | Aberrant morphological features in the cytoplasm of different erythrocyte variants pretreated with the indicated agents before infection with *Plasmodium falciparum*.** Tomograms of parasitized (top panels) and uninfected (bottom panels) HbAA erythrocytes pretreated with $H_2O_2$ (**a**), HbAC erythrocyte pretreated with DTT before infection (**b**) and HbAA erythrocyte pretreated with menadione (**c**). The dotted line in **c** indicates the edge of the EM carbon support. Colour code and labels as in Fig. 1. Representative examples of six biological replicates are shown for each case. Scale bars, 100 nm.

We found that the transient exposure to hydrogen peroxide or tert-butyl hydroperoxide raised the levels of methaemoglobin and irreversibly oxidized haemichromes to those found in HbAS, HbAC and HbF erythrocytes, as determined by Mössbauer spectroscopy (Table 1). Importantly, the *P. falciparum* strain FCR3[CSA] grew normally in pretreated cells, and both *var2csa* expression levels and haemozoin production were also normal (Fig. 2e,f). However, the amount of surface-presented VAR2CSA adhesin was significantly reduced (P<0.01; Fig. 2f). In addition, the ability of these cells to cytoadhere to CSA was diminished by ~70% compared with untreated controls (P<0.01 according to Kruskal–Wallis one-way analysis of variance test; Fig. 2d). Consistent with impaired cytoadherence, knobs were enlarged and low in numbers (Figs 2c and 3a and Supplementary Fig. 1). Furthermore, Maurer's clefts were malformed and the actin filaments were short and did not connect the knobs with the Maurer's clefts (Figs 2a,b and 3a and Supplementary Fig. 2). These data demonstrate that a transient oxidative insult can induce conditions in HbAA erythrocytes that resemble those found in haemoglobinopathic and fetal erythrocytes. The molecular species that interfere with actin reorganization and cytoadhesion seem to be irreversibly oxidized products, such as ferryl haemoglobin, haemichrome radicals or ferryl haem. This conclusion is supported by the finding that treating HbAC erythrocytes with 1 mM DTT for 60 min before infection could not reverse the intracellular environment to that of an HbAA erythrocyte; the levels of methaemoglobin and irreversibly oxidized haemichromes remained high (Table 1) and Maurer's cleft morphology (Figs 2b and 3b; and Supplementary Fig. 2 and Supplementary Table 1), knob density and shape (Figs 2c and 3b, and Supplementary Fig. 1), actin remodelling (Figs 2a and 3b) and cytoadhesion remained impaired (Fig. 2d). DTT itself

was inert in our assays and did not affect parasite growth, haemozoin production, or *var2csa* expression, as shown for HbAA erythrocytes (Fig. 2a–f).

**Menadione induces malaria-protective traits**. The finding that a transient oxidative stimulus could induce key features of the malaria-protective trait of haemoglobinopathic and fetal erythrocytes suggests new chemotherapeutic options against severe malaria. To explore this hypothesis, we selected menadione, a nutritional supplement and a precursor of vitamin K3 and a known oxidant of haemoglobin[37], for further investigation. Menadione induced the formation of methaemoglobin and irreversibly oxidized haemichromes in uninfected HbAA erythrocytes on a 60 min exposure at a concentration of 1 mM (Table 1). While the parasite developed normally in the menadione-pretreated erythrocytes (Fig. 2e), and although *var2csa* expression levels and haemozoin formation were not affected (Fig. 2e,f), parasites were unable to remodel the host actin, establish functional Maurer's clefts, form normal knobs and adhere efficiently (Figs 2a–f and 3c, and Supplementary Figs 1 and 2).

To further examine our hypothesis we tested the efficacy of pro-oxidative compounds in a rodent malaria model. To this end, C57BL/6 mice were subjected to three doses of 150 mg kg$^{-1}$ menadione sodium bisulphite administered intraperitoneally on 3 consecutive days (Fig. 4a). One day after the last treatment mice were infected intravenously with $10^6$ *Plasmodium berghei* ANKA-infected erythrocytes (Fig. 4a). This parasite line causes symptoms of experimental cerebral malaria during a normal course of infection[38]. Although the parasites multiplied at comparable rates in menadione-pretreated and untreated mice

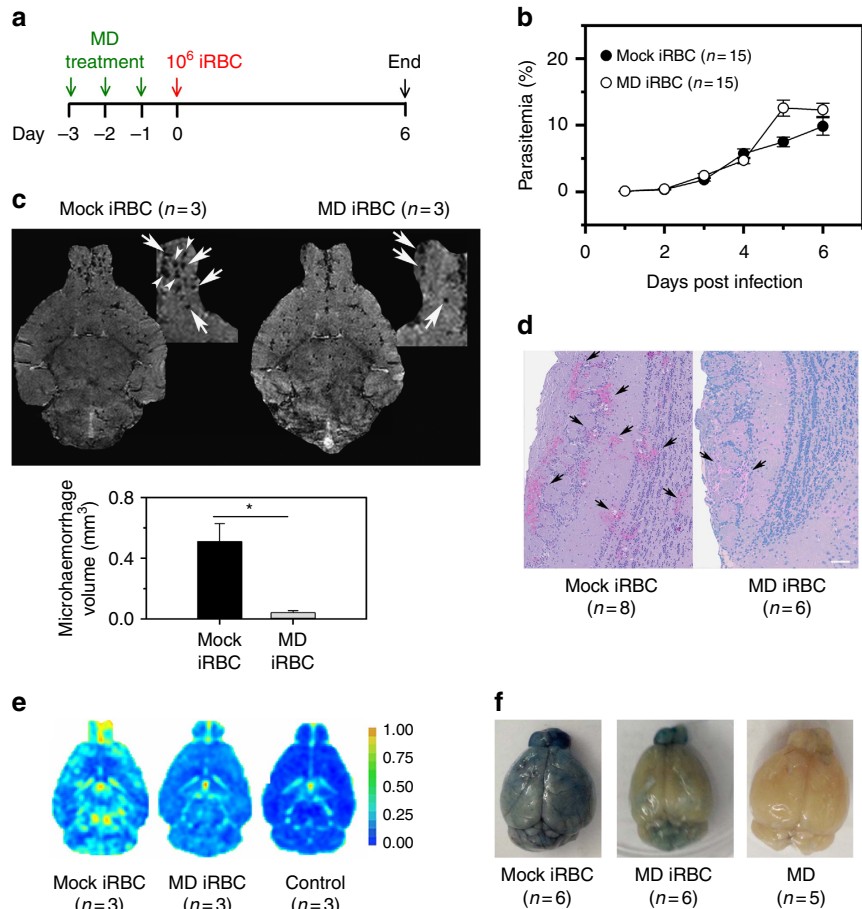

**Figure 4 | Effect of menadione pretreatment on *Plasmodium berghei* ANKA infections in mice.** (**a**) C57BL/6 mice were pretreated with intraperitoneal injections of 150 mg kg$^{-1}$ menadione (MD) or PBS (mock) on days $-3$, $-2$ and $-1$. On day 0 mice were intravenously infected with 10$^6$ *P. berghei*-infected erythrocytes (iRBC). One group of mice received only MD pretreatment and was not infected with parasites (control). The number of mice per treatment group is indicted in parenthesis. (**b**) Time course of parasitaemia in MD-pretreated and untreated mice. The mean ± s.e.m. of 15 mice per treatment group is shown. An *F*-test revealed no statistically significant differences between the two time courses. (**c**) Representative T2*-weighted magnetic resonance images of an MD-pretreated and an untreated *P. berghei*-infected mouse. The region of the olfactory bulb is magnified and microhaemorrhages are indicated by white arrows. Graph: quantification of the total volume of microhaemorrhages in the olfactory bulb. The mean ± s.e.m. of three mice per treatment group is shown ($P < 0.01$, according to Student's *t*-test). In this and in the following subfigures, mice were examined at day 6 post infection. (**d**) Histological examination of brain sections. Representative sections of the peripheral regions of the olfactory bulb, covering part of the periglomerular and mitral cell layer, from a MD-pretreated and an untreated infected mouse are shown. Note the reduced number of microhaemorrhages in the periglomerular layer (black arrows) from MD-pretreated infected mice compared with untreated infected mice. Scale bar, 100 μm (**e**) Representative T1-weighted subtraction magnetic resonance images after injection of a contrast agent, revealing less blood–brain barrier disruption in MD-pretreated infected mice as compared with untreated infected mice. A representative image of the brain of an uninfected and untreated mouse is shown as a negative control (control). The colour scale is a relative measure of blood–brain barrier disruption. (**f**) Vascular permeability of brains was further assessed by Evans Blue injection. Representative images of brains are shown for each group.

(Fig. 4b), the pretreated group developed significantly less brain damage (Fig. 4c–f). A key feature of *P. berghei*-induced brain damage is microhaemorrhages in the olfactory bulb[39,40]. These were less pronounced in the menadione-pretreated mice relative to untreated animals (microhaemorrhage volume of $0.042 \pm 0.015$ versus $0.51 \pm 0.12$ mm$^3$; $P = 0.018$ according to Student's *t*-test), as determined by magnetic resonance imaging of the brain at day 6 post infection and by histological examination of brain sections (Fig. 4c,d). In addition, we observed a trend towards a lesser degree of blood–brain barrier disruption in the menadione-pretreated group compared with the controls (average score $1.7 \pm 0.7$ and $2.7 \pm 0.3$, respectively), as assessed by magnetic resonance imaging and Evans Blue extravasation (Fig. 4e,f). These data suggest that exposure to the pro-oxidative drug menadione before infection can mitigate the development of brain damage in the *P. berghei* malaria model system.

## Discussion
We have recently shown that parasitized HbSC and HbCC erythrocytes display impaired host actin reorganization and malformed Maurer's clefts[10]. The present study extends these findings to parasitized HbAC and HbAS, and fetal erythrocytes (Figs 1 and 2). Other phenotypic aberrations associated with parasitized haemoglobinopathic and fetal erythrocytes include enlarged and dispersed knobs, aberrant presentation of adhesins (collectively termed *P. falciparum* erythrocyte membrane protein 1, PfEMP1) and reduced cytoadhesion (Figs 1 and 2)[2,3,9]. These additional phenotypic alterations have also been described in *ex vivo* field isolates[3,9]. It is therefore likely that our findings, although based on observations made using the long-term culture adapted *P. falciparum* clone FCR3$^{CSA}$, are representative. However, validation using *ex vivo* field isolates would be desirable. Given that the same aberrations occur in different

RBC variants and are independent of the parasite strain investigated and the adhesin variant expressed (see for comparison refs 2,3,9,10,22), this suggests a conserved mechanism. Our finding that a transient oxidative insult before infection can induce comparable phenotypes in parasitized HbAA erythrocytes points towards intracellular oxidative stress as the common underlying principle. This conclusion is further supported by the oxidative imbalance inherent to haemoglobinopathic and fetal erythrocytes (Table 1)[3,24,26,27,29].

On the basis of our findings we formulate the following model (Fig. 5): the malaria-protective trait of sickle cell haemoglobin, HbC and HbF results from a cascade of events that begins with the instability of these haemoglobin variants and a tendency to increased spontaneous auto-oxidation, which in turn leads to elevated quantities of irreversibly oxidized haemichromes, ferryl haemoglobin, protein radicals, free ferryl haem and other oxidants (Table 1)[3,24,26,27,29]. This pro-oxidative environment interferes with pathophysiological functions of the malaria parasite.

A parasite-induced process that is particularly vulnerable to oxidation is host actin remodelling (Fig. 5)[10]. It is known that actin is highly susceptible to oxidative stress and that oxidation impacts on actin polymerization and dynamics[10,30–32]. If host actin reorganization is impaired, then the actin network that connects the Maurer's clefts with the knobs cannot form and vesicular trafficking of adhesins to the host cell surface is compromised[13]. Moreover, Maurer's clefts do not properly develop because their morphology depends on a functional actin network, as previously shown for parasitized HbAA erythrocytes treated with cytochalasin D, an inhibitor of actin polymerization[10]. There is further circumstantial evidence that actin remodelling is required for knob formation.

The primary component of knobs is the knob-associated histidine-rich protein (KAHRP)[41]. KAHRP interacts with several parasite and host factors, including actin among other cytoskeletal proteins[42–49]. Interestingly, genetically engineered parasite lines encoding a truncated KAHRP protein in which part of the actin-binding domain has been deleted display knob phenotypes comparable to those seen in parasitized haemoglobinopathic and fetal erythrocytes[46]. Thus, the interaction of KAHRP with actin

seems to be crucial for proper knob assembly and the absence of this interaction might explain the presence of dispersed and enlarged knobs.

As a result of the impaired intracellular protein trafficking system, the amount of displayed adhesins is reduced, and those adhesins that are displayed are aberrantly presented as they are anchored in malformed and widely dispersed knobs (Fig. 5)[3,9]. Diminished cytoadherence would mitigate the life-threatening downstream sequellae resulting from the sequestration of parasitized erythrocytes in the microvasculature[12]. In our model, actin plays a central role as the primary target of reactive oxygen species from which all other events follow. While this is a plausible scenario due to the susceptibility of actin to oxidation, we cannot exclude the possibility of oxidative stress concurrently and independently impacting on several pathophysiological functions of the parasite. We further would like to counter the impression of oxidative stress being the sole mechanism by which HbS, HbC and HbF protect from severe malaria. Particularly in the case of HbS, there is evidence for multiple effector mechanisms, including inhibition of translation of parasite genes by host microRNAs, accelerated senescence and modulation of the host's immune response[5–7]. The presence of additional effector mechanisms might explain why HbAS is more protective in human populations than is HbAC[1].

Our findings regarding menadione provide a proof of concept for host-directed therapies that are designed to mimic the natural malaria-protective effects of HbS, HbC and HbF. However, we do not wish to advocate the use of systemically administered pro-oxidative drugs as antimalarial treatment. Systemic oxidative stress is a compounding disease factor of human malaria and further disturbing the redox balance might exacerbate suffering of patients[50]. Nevertheless, our data show that pretreatment of mice with menadione mitigated the progression of experimental cerebral malaria, resulting in fewer microhaemorrhages in the olfactory bulb and less blood–brain barrier damage, relative to untreated control mice. If oxidative drugs were considered as therapeutic options against severe malaria[51,52] then only compounds that locally exert their oxidative reactivity in the infected erythrocyte, possibly following activation of an inert prodrug.

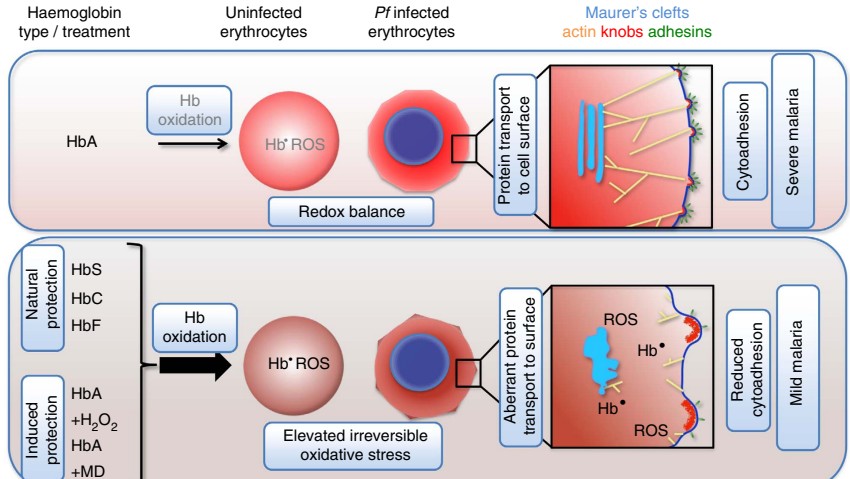

**Figure 5 | Model describing the proposed mechanism by which haemoglobinopathic erythrocytes protect from severe malaria.** The redox imbalance inherent to erythrocytes containing HbS, HbC or HbF leads to the production of reactive oxygen species (ROS), including globin-based radicals (Hb•). These ROS interfere with parasite-induced host actin reorganization and the generation of Maurer's clefts. As a consequence, delivery of adhesins to the host cell surface is hampered and the cytoadhesion complexes are malformed, with widely dispersed and enlarged knobs. In addition, adhesin molecules are aberrantly presented on the host cell surface and, thus, cytoadhesion is reduced. Pretreating wild-type HbAA erythrocytes with pro-oxidative agents, such as H₂O₂ or menadione (MD), causes oxidative stress, which, in turn, can induce the phenotype aberrations associated with protection from severe malaria by HbS, HbC and HbF.

We are aware that *P. berghei* ANKA-infected mice are a controversial surrogate disease system of human *P. falciparum* malaria since the respective pathogenic mechanisms might differ[38]. However, recent studies have shown that *P. berghei*-infected RBCs can sequester in different organs, including the brain[53,54], and that accumulation of parasitized RBCs, in addition to the recruitment and sequestration of activated CD8$^+$ T cells[55], is necessary for the development of cerebral malaria in mice[56]. Moreover, there is evidence of an evolutionary conserved machinery underlying sequestration and protein export to the host cell surface between *P. berghei* and *P. falciparum* species, although *P. berghei* lacks PfEMP1 adhesin orthologues[57]. Taking these findings into consideration, we explain the reduced brain damage in menadione-treated mice in light of our findings that menadione creates an oxidative imbalance in parasitized RBCs that interferes with the export and surface presentation of parasite-encoded virulence factors. In conclusion, our study reveals the causative role of an oxidative imbalance in erythrocytes containing HbS, HbC or HbF, which, in turn, prevents malaria parasites from establishing an efficient protein export machinery and a functional cytoadhesion complex, thus alleviating pathogenic interactions with the host.

## Methods

**Ethical clearance.** This study was approved by both the ethical review boards of Heidelberg University, Germany, and the Biomolecular Research Center (CERBA/Labiogene) at the University of Ouagadougou, Burkina Faso. Written informed consent was given by all blood donors. All animal experiments were performed according to the European regulations concerning FELASA category B and the GV-SOLAS standard guidelines. Animal experiments were approved by the German authorities (Regierungspräsidium Karlsruhe, Germany), § 8 Abs. 1 Tierschutzgesetz (TierSchG) under the license G-258/12 and were performed according to the National and European regulations. For all experiments, female C57BL/6 mice (8–10 weeks of age) were purchased from Janvier, France and were kept under specified pathogen-free conditions within the animal facility at Heidelberg University.

**Red blood cells.** Haemoglobinopathic erythrocytes (HbAC and HbAS) were donated in Burkina Faso and immediately shipped at 4 °C to Heidelberg University for further analysis. Fetal erythrocytes were collected at Heidelberg University Hospital and obtained from the umbilical cord following delivery. Haemoglobin genotypes were determined by PCR restriction fragment length polymorphism and cellulose acetate electrophoresis[4]. All cells were washed three times with cold AB medium (RPMI 1640 medium supplemented with 2 mM L-glutamine, 25 mM HEPES, 100 μM hypoxanthine and 20 μg ml$^{-1}$ gentamicin) and stored at 4 °C until used. All RBCs were used within 2 weeks after donation. Note that erythrocytes age at 4 °C *in vitro* and hence work on rapidly frozen *ex vivo* field isolates should be attempted to verify our results.

**Oxidative pretreatment of uninfected erythrocytes.** Uninfected HbAA erythrocytes were incubated with 1 mM H$_2$O$_2$ or tert-butyl hydroperoxide (tBOOH) for 1 h at room temperature[34]. After treatment, cells were washed three times with phosphate-buffered saline (PBS) before DTT was added to a final concentration of 1 mM to heal reversible oxidative damages[35]. Following treatment with menadione sodium bisulphite (1 mM for 1 h at room temperature, Sigma Aldrich), cells were washed five times. A volume of 3–4 ml of AB medium was added on top of the cell pellet after discarding the final wash. The pretreated erythrocytes were stored at 4 °C before use in parasite culture.

**Parasite strain.** The *P. falciparum* strain FCR3$^{CSA}$, preselected for high-capacity binding to chondroitin sulphate A (ref. 58), was used for all experiments.

**Culture of *P. falciparum*.** *P. falciparum* blood stages were cultured as described[59]. Briefly, cultures were maintained at a haematocrit of 5.0% and at a parasitaemia of not higher than 5% in RPMI 1640 supplemented with 2 mM L-glutamine, 25 mM HEPES, 100 μM hypoxanthine, 20 μg ml$^{-1}$ gentamycin, 5% human heat-inactivated AB-positive serum and 5% albumax. Cultures were incubated at 37 °C under controlled atmospheric conditions of 5% O$_2$, 3% CO$_2$, 92% N$_2$ and 96% humidity. To infect haemoglobinopathic or pretreated erythrocytes segmenters of the *P. falciparum* strain FCR3$^{CSA}$ cultured in wild-type erythrocytes were purified using a strong magnet (Miltenyi Biotech GmbH) as described elsewhere[60]. Briefly, the culture of ~3–5% trophozoites was applied to a MACS column attached to a 0.9 mm$^3$ 40 mm needle and allowed to flow through. The column was then washed with washing buffer (PBS supplemented with 0.5%

bovine serum albumin (BSA) and 2 mM EDTA) until the flow through was clear and the late-stage infected erythrocytes could be eluted. The eluted infected erythrocytes were subsequently washed twice in AB medium and used to inoculate a culture with a haematocrit of 5.0% of the appropriate RBC variants or of pretreated cells. Cultures were synchronized using 5% D-sorbitol and/or gelatin floatation[61,62]. The *P. falciparum* strain FCR3$^{CSA}$ was selected for the adhesive phenotype every 3 weeks by adhesion to plastic flasks coated with 10 mg ml$^{-1}$ CSA (Sigma Aldrich) as previously described[25]. Briefly, a 75 mm$^3$ sterile tissue culture flask (Cellstar, Greiner bio-one) was coated overnight with 14 ml 10 mg ml$^{-1}$ CSA. After washing with PBS, the flask was then coated with 1% BSA for 1 h at room temperature. Concurrently, enrichment of knob-infected late-stage erythrocytes was carried out by gelatin floatation and the resulting pellet resuspended in 14 ml of adhesion medium (RPMI 1640, 25 mM HEPES, pH 7.2). After the BSA blocking, the parasite was then added to the flask and incubated for 1 h at 37 °C. After removal of unbound infected erythrocytes by gentle washing 5–6 times with adhesion medium, the flask was washed strongly to detach CSA-bound cells, which were then used as inoculum for a new culture.

**Replication rate.** The growth phenotypes and replication rate for FCR3$^{CSA}$ cultured in various haemoglobinopathic and pretreated erythrocytes were assessed by Giemsa-stained blood smears over a period of at least 12 cycles (3–4 weeks).

**Adhesion assay.** Static adhesion assays were used to determine the adhesive behaviour of trophozoite-stage *P. falciparum* FCR3$^{CSA}$-infected erythrocytes to CSA (20–30 h post infection)[25,63]. Plastic Petri dishes were prepared by pre-coating several areas of 20 mm$^2$ overnight with 1 mg ml$^{-1}$ purified CSA or 1% BSA as a negative control. *P. falciparum* FCR3$^{CSA}$ were cultured with different haemoglobinopathic erythrocytes (HbAA, HbAS, HbAC and HbF) or pretreated erythrocytes for at least two cycles, and $5 \times 10^6$ infected erythrocytes (trophozoite stage) were applied onto these spots for 1 h. After multiple washing steps, the adherent cells were fixed with 2% glutaraldehyde for at least 2 h. After staining for 10 min with 10% Giemsa, three randomly selected areas from each spot were imaged using a Zeiss Axiovert 200M microscope (objective × 10/0.25 air differential interference contrast (DIC)). The amount of adhesive parasites were quantified using ImageJ as previously described[13]. Namely, the Giemsa-stained infected erythrocytes per image were automatically counted, using thresholding and watershed functions. All conditions are normalized as a percentage of infected HbAA erythrocytes in the same Petri dish.

**Expression levels of *var2csa*.** The *P. falciparum* strain FCR3$^{CSA}$ was grown in various haemoglobinopathic or pretreated erythrocytes. Total RNA was isolated from the cultures and cDNA synthesized as previously described[64]. Briefly, TRIzol and phenol/chloroform were used to extract total mRNA from ring stage cultures with 3–5% parasitaemia. A volume of 0.2 of chloroform was added per 1 volume TRIzol used and the upper aqueous layer was transferred to a new Eppendorf tube after 30 min of incubation at 4 °C. The aqueous layer was mixed with 0.5 volume of isopropanol and incubated for at least 2 h at 4 °C. To pellet the precipitated RNA, the sample was centrifuged at 13,000 r.p.m. for 1 h at 4 °C, the supernatant removed and ethanol added to wash the pellet. After air drying, the pellet was dissolved in distilled water and the RNA concentration measured. A unit of 2 μg of RNA was treated with DNAse and 100 μl taken for cDNA synthesis by RT–PCR. The sequence of the PCR primers used to quantify the amount of *var2csa* mRNA present in each sample can be found in Supplementary Table 2. The quantitative PCR reaction was performed using the StepOne qPCR machine (Applied Biosystems).

**Cryo-electron tomography.** Throughout this study intact, untreated and unfixed parasitized erythrocytes (at the trophozoite stage) were examined. Cryo-preparation and imaging was done as described[10]. Briefly, whole, intact parasitized erythrocytes were applied on to a holey EM grid and plunge frozen in liquid ethane cooled down with liquid nitrogen. The grids were investigated using a cryo-electron microscope (Polara, FEI, equipped with a field-emission gun and a Gatan GIF2002 post-column energy filter (Gatan), and operating at an accelerating voltage of 300 kV; FEI). The tomographic tilt series were recorded on a Gatan multi-scan charge-coupled device camera (Gatan) at × 18,000 nominal magnification, and typically consisted of 50–60 images collected in increments of 2° at a low electron dose of cumulatively 6,000–8,000 electrons per nm$^2$. Three-dimensional (3D) tomographic reconstructions were generated by weighted back-projection of aligned series of tilted images, using EM, TOM and IMOD image-processing software packages[10]. Resulting tomograms were analysed using the AMIRA volume-processing software (TGS Europe S.A.) for surface rendering and visualization. Actin filaments were tracked and their lengths were measured, using the AMIRA volume processing software (TGS Europe S.A.). Six tomograms were analysed for each condition.

**Scanning electron microscopy.** Infected erythrocytes were deposited on a coverslip and fixed with 2.5% glutaraldehyde + 1% osmium tetroxide for 1 h. The cells were then dehydrated in solutions containing increasing concentrations of acetone (30–100% in water) and dried using a critical point dryer. Alternatively, the dehydration was performed using an ethanol series (30–100% in water) followed by

soaking in 100% hexamethyldisalzan overnight. Dehydrated samples were mounted on studs and sputter-coated with 5–10 nm gold. The samples were viewed using a scanning electron microscope (Leo1530, Zeiss) at ×10,000 nominal magnification. At least 25 cells were investigated for each condition.

**Transmission electron microscopy.** Infected erythrocytes were prepared for transmission electron microscopy examination by high-pressure freezing—freeze substitution method as described[65]. The cells were frozen in a high-pressure freezer (HPF010, Bal-Tec). Freeze substitution was then performed in a Leica AFS freeze substitution unit (Leica). The 100 nm-thick sections were examined. Knob widths were measured at their bases in the plane of the surrounding plasma membrane. At least 25 cells were investigated for each condition.

**Measurement of volume and surface area of Maurer's clefts.** Samples of parasitized erythrocytes were prepared by High-Pressure Freezing/Freeze Substitution (Balzers HPM010; BAL-TECH AG, Lichtenstein). The 100 nm-thick sections were imaged using an electron microscope (FEI-CM120 Biotween, EMBL Heidelberg). For each experimental condition, we analysed data sets of more than 500 cell sections. The area of Maurer's clefts' membranous compartments (lamellae and vesicles) and the delineating borders were measured in EM 2D images, using both the AMIRA volume-processing package (TGS Europe S.A., France), and an established stereology method for surface and length measurements[66]. From this, the volume and membrane surface area (within 100 nm-thick sections) were calculated and extrapolated to the volume of a whole cell, using the Cavalieri principle[67]. The occurrence of Maurer's clefts extrapolated to an entire cell volume resulted in an average number of 12.3 Maurer's clefts per cell, which corresponds well with the observations made using fluorescence microscopy[19].

**Mössbauer spectroscopy.** Mössbauer spectroscopy was performed as described[68]. Briefly, packed erythrocytes were frozen in liquid nitrogen and mounted in a home-made cryostat. Mössbauer measurements were performed at a temperature of 85 K. The temperature was stabilized within ±0.1 K. The source of the 14.4 keV γ-radiation was 50 mCi 57Co(Rh). The velocity and the isomer shift were calibrated from the spectrum of α-iron foil measured at room temperature. Experimental data were analysed using the Recoil software[69].

**Quantification of haemozoin levels.** The amount of haemozoin was determined as previously described[70]. Briefly, infected erythrocytes at trophozoite and schizont stages were collected by centrifugation and washed twice with PBS. The pellet was then osmotically lysed by adding 50 ml of ice-cold distilled water (dH$_2$O) before centrifugation at 4,000 r.p.m. at 4 °C for 30 min to precipitate the haemozoin and the membrane ghosts. After the centrifugation, the white layer of RBC membrane ghosts was aspirated and the sticky black haemozoin pellet underneath washed twice with ice-cold dH$_2$O. These were then dissolved in 1 ml of 0.1 M NaOH and incubated at 50 °C for 10 min. The absorbance at 400 nm was used to determine the amount of haemozoin present; 100 μl of each sample was added to 96-well plates and a serial dilution of 0.1 M haemin chloride (Sigma Aldrich) was used to plot a standard curve. The concentration of haemozoin present in each sample was extrapolated from said graph.

**Determination of the level of VAR2CSA presentation.** The levels of VAR2CSA present was determined using flow cytometry, as previously described[71]. In short, infected erythrocytes at trophozoite stages (24–26 h post infection) were taken and washed with PBS supplemented with 2% fetal calf serum (FCS). A rabbit antiserum against baculovirus-produced recombinant domains of VAR2CSA was used. A volume of 3 μl of sera was used to label the samples for 30 min in a final volume of 50 μl PBS/FCS. After several washing steps, the samples were then labelled with 1:100 Alexa 488 goat anti-rabbit IgG (H + L) (Life Technologies) and 1:100 propidium iodide for 30 min in a final volume of 50 ml PBS/FCS. Uninfected erythrocytes were treated concurrently throughout as control. The fluorescence was finally determined using a FACScalibur (Becton Dickinson) and the CellQuest Pro Software 6.0.4 BD (Franklin Lakes).

**Mouse experiments.** Menadione sodium bisulphite (Sigma-Aldrich) was dissolved in 1× PBS and administered to groups of mice by intraperitoneal injection of 150 mg kg$^{-1}$ at days −3, −2 and −1. At day 0 mice were infected intravenously into the tail vein with a total of 10$^6$ *P. berghei* ANKA-infected erythrocytes. Parasitaemia was monitored daily by examination of Giemsa-stained blood smears. At day 6 post infection mice were examined for brain damage. Magnetic resonance imaging of the brain and data analysis were performed as previously described[39]. Briefly, mice were anaesthetized by isoflurane inhalation at day 5 post infection and placed prone in fixed position. Magnetic resonance imaging scans were performed on a 9.4T small animal scanner (BioSpec 94/20 USR, Bruker Biospin GmbH, Ettlingen, Germany) using a volume resonator for transmission and a 4-channel-phased array surface receiver coil. Image processing was conducted in Amira 5.4 (FEI, Visualization Sciences Group).

Microhaemorrhage volume was assessed semi-automatically on T2*-weighted data sets in the olfactory bulb (T2*-weighted flow compensated gradient echo imaging; TR/TE = 50/18 ms, FA = 12°, 80 μm isotropic resolution). Blood–brain barrier permeability was analysed by contrast-enhanced T1w imaging (3D T1-weighted imaging pre- and post-contrast agent injected of Gd-DTPA at a dosage of 0.3 mmol kg$^{-1}$; TR/TE = 5/1.9 ms, FA = 8.5°, 156 μm isotropic resolution). 3D non-enhanced T1w images were subtracted from enhanced T1w images; pathological enhancement of difference images was graded into mild (= 1), moderate (= 2) and severe (= 3). Cerebral vascular leakage was investigated using Evans Blue[72]. Histological examinations of mouse brain sections was performed as described[73]. Briefly, Mice were killed and perfused with 10 ml PBS and 10 ml 4% paraformaldehyde before brains were carefully removed and incubated in 4% paraformaldehyde for 24 h at 37 °C. Brains were snap frozen in liquid nitrogen. Ten-micrometre-thick coronal sections were cut on a cryotome (Leica Microsystems, Vienne, Austria) and directly mounted onto SuperFrost/Plus slides (Microm International, Walldorf, Germany). Serial sections were stained routinely with haematoxylin and eosin (Thermo Fisher Scientific, Waltham, MA) and Giemsa (Thermo Fisher Scientific). Slides were blinded, and three randomly selected sections per coronal plane were subjected to histological analysis. Examinations were performed on an Olympus BX45 research microscope (Olympus, Tokyo, Japan). Histological sections of organs were examined in a blind manner.

**Statistical analyses.** Statistical significance was assessed using Sigma Plot 13 and the Kruskal–Wallis one-way analysis of variance test or, where appropriate, the Student's *t*-test.

**Data availability.** The authors declare that the data supporting the findings of this study are available within the article and its Supplementary Information files, or available from the authors on request.

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

## Acknowledgements

The VAR2CSA antiserum was a kind gift from Ali Salanti (University of Copenhagen). We thank Rebecca Compaore (University of Ouagadougou) for collecting some of the blood samples. We are grateful to John Briggs and Yannick Schwab (EMBL) for access to the EM core facility at the EMBL, Heidelberg. We acknowledge Jozef Korecki (Kraków University of Science and Technology) for kindly providing us with the Co(Rh) source. We thank Marcel Deponte and Ross Douglas (Heidelberg University Medical Hospital) for critical discussions. This work was in part supported by the Deutsche Forschungsgemeinschaft under the Collaborative Research Center SFB 1129 (M.L., A.-K.M. and F.F.), the European Research Council (StG 281719; F.F.), the Bill and Melinda Gates Foundation (Grand Challenges Explorations phase I; Nr: OPP1069409; M.C.) and the German Center for Infection Research (DZIF; F.L., M.L. and A.-K.M.). M.L.and F.F. are members of the research cluster of excellence CellNetworks, and S.S. and B.N. are members of the HBIGS graduate school at Heidelberg University. We thank the electron microscopy core facility of Heidelberg University for help with, and access to, their microscopes.

## Author contributions

M.C., S.S., B.N., K.B., S.A., C.B., A.H., F.L. and C.P.S. performed experiments and analysed data; M.C., J.S., F.F., A.-K.M, C.P.S. and M.L. designed experiments and analysed data; M.L. and M.C. wrote the manuscript and designed the figures. All authors discussed results and commented on the manuscript.

## Additional information

**Competing financial interests:** The authors declare no competing financial interests.

