## [Peer Review File · Nature Communications]

Reviewers' comments:

Reviewer #1 (Remarks to the Author):

Lines 51-52, The phrase "...from the Maurer's clefts to knob-like protrusions..." and the cyan coloration in Figure 1 may give readers the impression that the vesicles actually derive from the Maurer's clefts. This possibility seems intuitive, but has it been confirmed? If not, what is the source of these vesicles?

Line 57, The observations of enlarged and dispersed knobs were also nicely documented and quantified in Arie et al., *J Struct Biol*, 2005. Consider citing here and in line 80 so that readers can compare knob densities and sizes between this previous report and the present manuscript.

Lines 62-64 and Figure 1, One gets the impression that the vesicles are transporting PfEMP1 from Maurer's clefts to knobs along polymerized actin filaments; that is, the knobs are already in place as the destination for the vesicles. Do the authors intend to give this impression? If so, then it would seem that knobs are pre-formed by a different mechanism. This is an important distinction to make since it would imply that other major knob components such as KAHRP and PfEMP3 are not transported from Maurer's clefts to knob assembly sites via polymerized actin filaments. It would logically follow that the lack of long actin filaments is not the cause of abnormal knob formation. This potential interpretation is apparent in Figures 1b and 1c, where knobs are present at the surface of parasitized HbAS and HbAC erythrocytes despite the presence of unattached actin filaments. Can the authors discuss these points?

Line 66, The authors have exclusively used a single long-term-adapted parasite line (FCR3-CSA) in this study. Have they confirmed any of their fundamental observations using parasite clinical isolates *ex vivo*? If not, this may be mentioned as a limitation of this study, which can be addressed in future work.

Figures 1a and 1b, The reconstructed volume images of uninfected HbAA and HbAS erythrocytes seem to show qualitative differences in the density and organization of short actin filaments. Assuming these are representative images, can the authors comment on these differences, and also on whether they also observed them in uninfected HbAC and HbF erythrocytes?

Figures 1c and 1d, The tomograms and reconstructed volume images of uninfected HbAC and HbF erythrocytes are conspicuously absent from these panels and should probably be included for completeness.

Figure 2a, Is there any reason the authors did not quantify the actin length in uninfected HbAC and HbF erythrocytes for comparison to their infected counterparts?

In Figure 2 (all five panels) and Table 1, there are no suggestions that HbAS confers a more dramatic phenotype than HbAC, and yet HbAS is more protective in human populations than

HbAC. Can the authors speculate about this apparent disconnect between phenotype and protection?

Lines 91-93, Consider mentioning here that these observations are consistent with those made previously for other PfEMP1 variants (references 2, 3, and 5).

Line 98, Suggest replacing "accidental" with "spontaneous," since erythrocytes have probably evolved this mechanism to ensure their eventual mechanical clearance in the spleen.

Line 107, Consider mentioning here that elevated levels of hemichromes were previously observed and associated with the abnormal knob phenotype in HbAC and HbCC erythrocytes (reference 3).

Lines 122-124, How did the authors select the 60 minute duration and 1 mM oxidant and reducing agent concentrations for these experiments?

Line 137, Edit "2b" to "2a."

Line 183, Were cord blood samples also obtained in Burkina and shipped to Heidelberg?

Line 188, Since erythrocytes age at 4C in vitro, and HbS, HbC, and HbF erythrocytes age more quickly than HbA erythrocytes, the authors should mention that the 2-week age-range of erythrocytes is a potential limitation of this study. That is, some of their observations may be less dramatic than if infected or uninfected erythrocytes were obtained from donors and used immediately in assays.

Figures 3b and 3c, Consider adding the tomograms and reconstructed volume images of uninfected HbAC/DTT and HbAA/menadione erythrocytes for completeness.

Reviewer #2 (Remarks to the Author):

Previously the authors have reported abnormal findings with the actin cytoskeleton and the Maurer's clefts in erythrocytes from hemoglobin S and C. Here, in this manuscript Cyrklaff et al. have reported a mechanism by which erythrocytes undergo a variety of alterations in cytoskeletons including actin and others. According to their findings, they assumed that redox imbalance, which is seen with erythrocytes from sickle cell disease and fetal erythrocytes, plays a critical role as the initial molecular event in the cascade of molecular events including cytoskeletons. The authors observed similar molecular abnormalities in erythrocytes from normal individuals to which they gave a transient oxidative stress by menadione. Thus, they concluded that oxidative stress can be used for possible interventions for malaria infection.

Oxidative stress associated with malaria infection has been well documented thus far and is not a novel finding; there are many papers describing the presence of oxidative stress in malarial infection. It is not clear whether the findings on redox imbalance in malaria infection is a novel finding. Also, under a clinical condition with excessive oxidative stress, it is not certain to use pro-oxidative compounds for therapeutic purposes, as it would be highly possible to aggravate patients' clinical symptoms or the pathophysiology relevant to malaria infection. Further the clinical use of menadione in populations that are at risk of malaria infection is not validated yet because of multiple adverse effects on erythrocytes.

The molecular techniques that were employed in this manuscript are quite impressive in terms of the presentation of current status of multiple molecules in erythrocytes involved in malaria infection. The quality of the data presented here are of great value for the understanding molecular events in malaria. The experimental procedures for the measurements of oxy-hemoglobin, methhemoglobin, and oxidized hemoglobin are not immediately clear in this manuscript. How many samples were measured for each variant? There are no such data on these results.

Although the findings presented here with the dissection of molecules inside erythrocytes look very interesting, the roles of oxidative stress in malaria infections are well known. The potentially protective roles of such oxidative stress associated with malaria infection was already reported before. In contrast, some paper has suggested a possible application of antioxidant to prohibit from cerebral malaria. The conclusions drawn in this manuscript may not be supported by the experimental evidence shown here. The roles of pro-oxidant use for treating malaria infection should be tested in vivo using a mouse model but not in vitro systems only.

Some in vivo studies should be performed to confirm the validity of the findings shown here, rather than in vitro studies only.

Reviewer #3 (Remarks to the Author):

This work uses electron tomography (ET) of vitreously frozen samples to examine the membrane skeleton of human red blood cells (RBCs) of normal, heterozygous HbAC or HbAS and fetal origin infected with *P. falciparum*-infected. The authors examined alterations induced by the parasite on these different host cell background. In HbAC, HbAS and HbF backgrounds the authors report aberrant remodelling of host actin remodelling, impaired presentation of surface adhesins, and reduced cytoadherence. These changes are associated with the presence of irreversibly oxidized haemoglobin. They further report that oxidative insult to normal RBCs prior to infection induces irreversible (oxidative) changes leading to similar phenotypic features to those observed in HbAC, HbAS and HbF RBCs. Of particular interest the authors found that treatment of RBCs with menadione induced a similar effect. This is an elegant study with interesting findings, but there are some points that should be addressed before this work is accepted for publication.

Figure 1 illustrates structural features at the host RBC membrane in RBCs with different

haemoglobin variants infected with trophozoites. The organisation of the RBC membrane skeleton appears to be different in HbAA and HbAS RBCs. This is evident as a wider distribution of actin filament lengths in the HbAS RBCs in Figure 2A. Equivalent data should be shown for HbAC and HbF uninfected RBCs and these findings should be discussed.

Assessment of the lengths of the actin filaments provides good evidence that the difference in membrane skeleton organisation are statistically significant. The authors also claim that the FCR3 parasites do not establish lamellar Maurer's clefts in the aberrant RBCs. There is however no quantification of this effect and it is not possible to make an assessment of Maurer's cleft structures on the basis of the very limited sample sets provided. If the Maurer's clefts are no longer tethered to the membrane skeleton they may be present elsewhere in the RBC but still lamellar in structure. To address this question it would be important to use another lower resolution technique that is able to assess the distribution and organisation of Maurer's clefts in the aberrant RBC on a scale that permits statistical analyses. This could employ high resolution light microscopy or a higher-throughput electron microscopy technique, such as block-face SEM.

The authors state that cytoadhesion remained impaired following treatment of RBCs with 1 mM DTT for 60 min prior to infection. However there appears to be a marked increase in adhesion to CSA upon treatment of HbAA and HbAC RBCs prior to infection (Fig 2C). If there is no increase in surface Var2csa and no other evident changes the authors need to address the reason for this remarkable enhancement of adhesion.

Minor Points: The filament observed in the uninfected RBC membrane skeleton are referred to as actin filaments. Is there any evidence that these filaments are comprised solely of actin? It would seem more likely that they are spectrin cross-members linking into actin junction points, as imaged in previous cryoEM studies [1].

1. Nans, A., N. Mohandas, and D.L. Stokes, Native ultrastructure of the red cell cytoskeleton by cryo-electron tomography. *Biophys J*, 2011. 101(10): p. 2341-50.

An oxidative insult can induce the malaria-protective trait of sickle and foetal erythrocytes

We thank the reviewers for their positive comments and helpful suggestions. Detailed responses to the reviewer's comments are set out below.

Reviewer: 1

Comment 1: Lines 51-52, The phrase "...from the Maurer's clefts to knob-like protrusions..." and the cyan coloration in Figure 1 may give readers the impression that the vesicles actually derive from the Maurer's clefts. This possibility seems intuitive, but has it been confirmed? If not, what is the source of these vesicles?

Reply: There is independent evidence from Leann Tilley's and my group, using cryo-electron tomographic techniques and confocal laser scanning microscopy, that PfEMP1-carrying vesicles bud off from the Maurer's clefts. There is further evidence of vesicles originating from the parasitophorous vacuolar membrane. To clarify this issue we added the following sentence to the manuscript:

"In addition to vesicles budding off from the Maurer's clefts, there is evidence of transport vesicles originating from the parasitophorous vacuolar membrane that separates the parasite from the host erythrocyte ^{10, 17, 18}." (introduction: page 3; last sentence, and following page).

We further added the appropriate references.

Comment 2: Line 57, The observations of enlarged and dispersed knobs were also nicely documented and quantified in Arie et al., J Struct Biol, 2005. Consider citing here and in line 80 so that readers can compare knob densities and sizes between this previous report and the present manuscript.

Reply: The reference has been added (reference 20 in the manuscript).

Comment 3: Lines 62-64 and Figure 1, One gets the impression that the vesicles are transporting PfEMP1 from Maurer's clefts to knobs along polymerized actin filaments; that is, the knobs are already in place as the destination for the vesicles. Do the authors intend to give this impression? If so, then it would seem that knobs are pre-formed by a different mechanism. This is an important distinction to make since it would imply that other major knob components such as KAHRP and PfEMP3 are not transported from Maurer's clefts to knob assembly sites via polymerized actin filaments. It would logically follow that the lack of long actin filaments is not the cause of abnormal knob formation. This potential interpretation is apparent in Figures 1b and 1c, where knobs are present at the surface of parasitized HbAS and HbAC erythrocytes despite the presence of unattached actin filaments. Can the authors discuss these points?

Reply: The reviewer raises an interesting issue. However, we are not sure if there is a final answer to the question. According to our current understanding, KAHRP, the main component of knobs, is exported into the erythrocyte cytosol as a soluble protein, where it spontaneously associates with PfEMP1 at Maurer's clefts to form the cytoadhesion complex (Wickham, et al. 2001 Embo J 20, 5636-5649). KAHRP further binds to membrane skeletal components and it self-aggregates, which explains the formation of knobs below the inner leaflet of the erythrocyte's plasma membrane (references 42-49 in the manuscript). Thus, there is no evidence of actin-mediated vesicular transport playing a role in trafficking of KAHRP to the erythrocyte surface. Consistent with these observations, knobs are present in parasitized haemoglobinopathic erythrocytes, in spite of impaired host actin remodeling, as shown in several studies (references 2, 3, 9, and 20 in the manuscript). However, they are

aberrant – dispersed and inflated, suggesting that a critical component for proper assembly is missing. We speculate that the missing link is the interaction of KAHRP with actin. This hypothesis is based on the following observations: i) KAHRP contains an actin binding domain. ii) Genetically engineered parasite lines expressing a truncated KAHRP protein lacking the actin binding domain displayed a knob phenotype very similar to that seen in parasitized haemoglobinopathic erythrocytes – dispersed and enlarged knobs. While actin is not required for KAHRP trafficking, actin and actin reorganization might play a role in knob assembly. Alternatively or in addition, KAHRP-actin interactions may play a role in limiting knob assembly to an optimal size or play a role in distribution of knobs on the surface. Clearly, much more work will be needed to fully understand knob formation. To clarify this issue we have added the following paragraph to the manuscript:

“There is further circumstantial evidence that actin remodeling is required for knob formation. The primary component of knobs is the knob-associated histidine-rich protein (KAHRP)⁴¹. KAHRP interacts with several parasite and host factors, including actin among other cytoskeletal proteins 42-49. Interestingly, genetically engineered parasite lines encoding a truncated KAHRP protein in which part of the actin binding domain has been deleted display knob phenotypes comparable to those seen in parasitized haemoglobinopathic and foetal erythrocytes 46. Thus, the interaction of KAHRP with actin seems to be crucial for proper knob assembly and the absence of this interaction might explain the presence of dispersed and enlarged knobs.” (discussion: page 11, end of first paragraph and following paragraph)

Comment 4: Line 66, The authors have exclusively used a single long-term-adapted parasite line (FCR3-CSA) in this study. Have they confirmed any of their fundamental observations using parasite clinical isolates ex vivo? If not, this may be mentioned as a limitation of this study, which can be addressed in future work.

Reply: While not formally verified, several of the phenotypic alterations described in this study are consistent with observations made using ex vivo field isolates or uninfected erythrocytes. This includes oxidative stress and a looser organized membrane skeleton in HbAS, HbAC, and HbF erythrocytes (references 3, 24, 26, and 27 in the manuscript) and aberrant knob formation and impaired cytoadhesion in ex vivo field isolates from carriers of HbAS and HbAC (references 2, 3, and 9 in the manuscript). Moreover, we made the same observation of actin filaments linking Maurer’s clefts to knobs in three different culture adapted parasite lines (references 10 and 22 in the manuscript). This is now also mentioned at the start of the We have amended the text as following:

“These additional phenotypic alterations have also been described in ex vivo field isolates^{3,9}. It is therefore likely that our findings, although based on observations made using the long-term culture adapted *P. falciparum* clone FCR3^{CSA}, are representative. However, validation using ex vivo field isolates would be desirable. Given that the same aberrations occur in different red blood cell variants and are independent of the parasite strain investigated and the adhesin variant expressed (see for comparison^{2, 3, 9, 10, 22}), this suggests a conserved mechanism.” (discussion: page 10, middle of first paragraph).

Comment 5: Figures 1a and 1b, The reconstructed volume images of uninfected HbAA and HbAS erythrocytes seem to show qualitative differences in the density and organization of short actin filaments. Assuming these are representative images, can the authors comment on these differences, and also on whether they also observed them in uninfected HbAC and HbF erythrocytes?

Reply: We thank the reviewer for pointing this out to us. We now have quantified the differences in these uninfected cells and found that the organization of actin indeed differs between uninfected HbAA and HbAS, HbAC and HbF erythrocytes. The actin filaments are significantly longer in HbAS, HbAC and HbF erythrocytes relative to those in HbAA

erythrocytes (see revised Figure 2a for quantitative measurements of actin length in uninfected erythrocytes). This finding is consistent with a dysfunctional cytoskeleton, as has been demonstrated for HbAS erythrocytes (references 24 in the manuscript). To clarify this issue we have added the following sentences to the manuscript:

“When comparing the actin length of uninfected erythrocytes, we noted a statistically significant difference between HbAA on the one hand and HbAS, HbAC and HbF red blood cells on the other hand ($p < 0.01$; Fig. 2a), consistent with a looser organized membrane skeleton in these cells, as has been demonstrated for uninfected HbAS erythrocytes²⁴.” (results: page 5, end of first paragraph).

Comment 6: Figures 1c and 1d, The tomograms and reconstructed volume images of uninfected HbAC and HbF erythrocytes are conspicuously absent from these panels and should probably be included for completeness.

Reply: We apologize for this oversight. Tomographic images of uninfected HbAC and HbF are now shown in the revised Figure 1. From the tomograms, we obtained quantitative data on actin length, which are now presented in the revised Figure 2a. These additional data are described in the results section on page 5, end of first paragraph.

Comment 7: Figure 2a, Is there any reason the authors did not quantify the actin length in uninfected HbAC and HbF erythrocytes for comparison to their infected counterparts?

Reply: We apologize for this oversight. The actin length in uninfected HbAA, HbAS, HbAC, and HbF has been characterized and compared to their infected counterparts. The data are now presented in the revised Figure 2a and are described in the results section on page 5, second paragraph. See also answers to the previous two comments.

Comment 8: In Figure 2 (all five panels) and Table 1, there are no suggestions that HbAS confers a more dramatic phenotype than HbAC, and yet HbAS is more protective in human populations than HbAC. Can the authors speculate about this apparent disconnect between phenotype and protection?

Reply: The reviewer raises an interesting question to which there is currently no definitive answer given our limited understanding of the underlying processes. In the manuscript, we now speculate that effector mechanisms, in addition to oxidative stress, might contribute to the malaria protective function of HbAS. To clarify this issue we have added the following sentences to the manuscript:

“We further would like to counter the impression of oxidative stress being the sole mechanism by which haemoglobin S, C, and F protect from severe malaria. Particularly in the case of haemoglobin S, there is evidence for multiple effector mechanisms, including inhibition of translation of parasite genes by host microRNAs, accelerated senescence, and modulation of the host’s immune response 5-7. The presence of additional effector mechanisms might explain why HbAS is more protective in human populations than is HbAC¹.” (discussion: page 11, end of last paragraph).

Comment 9: Lines 91-93, Consider mentioning here that these observations are consistent with those made previously for other PfEMP1 variants (references 2, 3, and 5).

Reply: We thank the reviewer for pointing this out to us. The following sentence has been added to the manuscript:

“Given that the same aberrations occur in different red blood cell variants and are independent of the parasite strain investigated and the adhesin variant expressed (see for

comparison^{2, 3, 9}), this suggests a conserved mechanism." (discussion, page 10, middle of first paragraph).

Comment 10: Line 98, Suggest replacing "accidental" with "spontaneous," since erythrocytes have probably evolved this mechanism to ensure their eventual mechanical clearance in the spleen.

Reply: Done as suggested. (results: page 6, middle of third paragraph).

Comment 11: Line 107, Consider mentioning here that elevated levels of hemichromes were previously observed and associated with the abnormal knob phenotype in HbAC and HbCC erythrocytes (reference 3).

Reply: Done as suggested. (results: page 7, end of first paragraph).

Comment 12: Lines 122-124, How did the authors select the 60 minute duration and 1 mM oxidant and reducing agent concentrations for these experiments?

Reply: We followed established protocols to induce oxidative stress in erythrocytes. The protocols are now cited in the manuscript (references 34 and 34).

Comment 13: Line 137, Edit "2b" to "2a."

Reply: Done as suggested.

Comment 14: Line 183, Were cord blood samples also obtained in Burkina and shipped to Heidelberg?

Reply: Cord blood was obtained from maternity ward at Heidelberg University Hospital and immediately used in infection experiments. We clarified this issue in the methods section (page 13, second paragraph).

Comment 15: Line 188, Since erythrocytes age at 4C in vitro, and HbS, HbC, and HbF erythrocytes age more quickly than HbA erythrocytes, the authors should mention that the 2-week age-range of erythrocytes is a potential limitation of this study. That is, some of their observations may be less dramatic than if infected or uninfected erythrocytes were obtained from donors and used immediately in assays.

Reply: This is certainly a possibility. We acknowledge this fact and now state in the manuscript:

"..... validation (of our observations) using ex vivo field isolates would be desirable." (discussion: page 10, middle of first paragraph).

and

"Note that erythrocytes age at 4°C in vitro and hence work on rapidly frozen ex vivo field isolates should be attempted to verify our results." (methods: page 13, end of second paragraph).

Comment 16: Figures 3b and 3c, Consider adding the tomograms and reconstructed volume images of uninfected HbAC/DTT and HbAA/menadione erythrocytes for completeness.

Reply: The requested tomograms are now shown in the revised Figure 3.

Reviewer: 2

Previously the authors have reported abnormal findings with the actin cytoskeleton and the Maurer's clefts in erythrocytes from hemoglobin S and C. Here, in this manuscript Cyrklaff et al. have reported a mechanism by which erythrocytes undergo a variety of alterations in cytoskeletons including actin and others. According to their findings, they assumed that redox imbalance, which is seen with erythrocytes from sickle cell disease and fetal erythrocytes, plays a critical role as the initial molecular event in the cascade of molecular events including cytoskeletons. The authors observed similar molecular abnormalities in erythrocytes from normal individuals to which they gave an transient oxidative stress by menadione. Thus, they concluded that oxidative stress can be used for possible interventions for malaria infection.

Oxidative stress associated with malaria infection has been well documented thus far and is not a novel finding; there are many papers describing the presence of oxidative stress in malarial infection. It is not clear whether the findings on redox imbalance in malaria infection is a novel finding. Also, under a clinical condition with excessive oxidative stress, it is not certain to use pro-oxidative compounds for therapeutic purposes, as it would be highly possible to aggravate patients' clinical symptoms or the pathophysiology relevant to malaria infection. Further the clinical use of menadione in populations that are at risk of malaria infection is not validated yet because of multiple adverse effects on erythrocytes.

The molecular techniques that were employed in this manuscript are quite impressive in terms of the presentation of current status of multiple molecules in erythrocytes involved in malaria infection. The quality of the data presented here are of great value for the understanding molecular events in malaria.

Comment 1: The experimental procedures for the measurements of oxy-hemoglobin, methhemoglobin, and oxidized hemoglobin are not immediately clear in this manuscript. How many samples were measured for each variant? There are no such data on these results.

Reply: We apologize for this oversight. To clarify this issue we added the following sentence to the legend of Table 1:

“The experimental uncertainty of the determinations is less than 1.5% and the mean of three independent biological replicates, using blood from three different donors, is shown.” (Table 1 legend: page 26)

Comment 2: Although the findings presented here with the dissection of molecules inside erythrocytes look very interesting, the roles of oxidative stress in malaria infections are well known. The potentially protective roles of such oxidative stress associated with malaria infection was already reported before. In contrast, some paper has suggested a possible application of antioxidant to prohibit from cerebral malaria. The conclusions drawn in this manuscript may not be supported by the experimental evidence shown here. The roles of pro-oxidant use for treating malaria infection should be tested in vivo using a mouse model but not in nitro systems only. Some in vivo studies should be performed to confirm the validity of the findings shown here, rather than in vivo studies only.

Reply: We thank the reviewer for the insightful comments and suggestion. We have now added an entire figure (Figure 4) on in vivo data using a mouse model (see below). The reviewer is perfectly right in saying that oxidative stress is a compounding disease factor of malaria (we acknowledge this now in the revised manuscript (discussion page 12; first paragraph). The reviewer is also right in stating that menadione has adverse effects that preclude the drug from being used in the treatment of malaria patients who already suffer from a redox imbalance. We agree with the reviewer on all these issues and apologize if

some ambiguous phrases in the text mislead the reviewer in assuming otherwise. We hope to have now made the focus of the manuscript clearer.

In the present manuscript, we attempted to address a fundamental biological question, namely how haemoglobinopathic and foetal red blood cells protect from severe malaria. The purpose of using menadione in our study has been to demonstrate proof-of-concept for host-directed therapies that mimic the natural malaria protective effects of HbS, HbC, and HbF. We can clearly show that menadione and other oxidative agents can induce features of the malaria protective effects in wild type HbAA erythrocytes under in vitro conditions (Figures 2 and 3). In the revised manuscript, we have added an entire figure showing data from experiments where we test this finding in a rodent malaria model. We found that mice pretreated intraperitoneally with menadione prior to infection developed significantly less brain damage relative to untreated mice. These important new findings are presented in the new Figure 4 and are described in the results section on page 9 (second paragraph) and are discussed on page 12, first and second paragraph.

Reviewer: 3

This work uses electron tomography (ET) of vitreously frozen samples to examine the membrane skeleton of human red blood cells (RBCs) of normal, heterozygous HbAC or HbAS and fetal origin infected with P. falciparum-infected. The authors examined alterations induced by the parasite on these different host cell background. In HbAC, HbAS and HbF backgrounds the authors report aberrant remodelling of host actin remodelling, impaired presentation of surface adhesins, and reduced cytoadherence. These changes are associated with the presence of irreversibly oxidized haemoglobin. They further report that oxidative insult to normal RBCs prior to infection induces irreversible (oxidative) changes leading to similar phenotypic features to those observed in HbAC, HbAS and HbF RBCs. Of particular interest the authors found that treatment of RBCs with menadione induced a similar effect. This is an elegant study with interesting findings, but there are some points that should be addressed before this work is accepted for publication.

Comment 1: Figure 1 illustrates structural features at the host RBC membrane in RBCs with different haemoglobin variants infected with trophozoites. The organisation of the RBC membrane skeleton appears to be different in HbAA and HbAS RBCs. This is evident as a wider distribution of actin filament lengths in the HbAS RBCs in Figure 2A. Equivalent data should be shown for HbAC and HbF uninfected RBCs and these findings should be discussed.

Reply: We thank the reviewer for pointing out these shortcomings in our manuscript. We now show tomograms for both infected erythrocytes and their uninfected counterparts (revised Figures 1 and 3) and we have further quantified and compared the length of the actin filaments in the different infected and uninfected red blood cell (Figure 2a) . This indeed revealed a quantitative difference in filament length. Please see our reply to comments 5, 6, 7, and 16 made by reviewer 1.

Comment 2: Assessment of the lengths of the actin filaments provides good evidence that the difference in membrane skeleton organisation are statistically significant. The authors also claim that the FCR3 parasites do not establish lamellar Maurer's clefts in the aberrant RBCs. There is however no quantification of this effect and it is not possible to make an assessment of Maurer's cleft structures on the basis of the very limited sample sets provided. If the Maurer's clefts are no longer tethered to the membrane skeleton they may be present elsewhere in the RBC but still lamellar in structure. To address this question it would be important to use another lower resolution technique that is able to assess the distribution and organisation of Maurer's clefts in the aberrant RBC on a scale that permits statistical

analyses. This could employ high resolution light microscopy or a higher-throughput electron microscopy technique, such as block-face SEM.

Reply: We now provide extensive semi-quantitative data on the volume and surface area of Maurer's clefts (new Figure 2b and Extended Data Figure 2). These new data suggest that the atypical vesicular morphology of the Maurer's clefts results from a volume expansion at constant surface area, i.e., the Maurer's clefts change their shape from a fattened lamellar profile with a low volume to surface area ratio to a vesicular form. These new findings are described in the results section on page 5, end of the last paragraph.

Comment 3: The authors state that cytoadhesion remained impaired following treatment of RBCs with 1 mM DTT for 60 min prior to infection. However there appears to be a marked increase in adhesion to CSA upon treatment of HbAA and HbAC RBCs prior to infection (Fig 2C). If there is no increase in surface Var2csa and no other evident changes the authors need to address the reason for this remarkable enhancement of adhesion.

Reply: We carefully reanalyzed the data presented in the Figure and found no statistical significance between DTT pre-treated and untreated parasitized HbAC erythrocytes (Kruskal-Wallis One Way ANOVA on Ranks test). In this context, we regrettably have to admit that a mistake happened in the labeling of the original figure as the labels for iHbF and iHbAC-DTT were mixed up. We have corrected the error in the revised figure and thank the reviewer for helping us correct this mistake.

Comment 4: Minor Points: The filament observed in the uninfected RBC membrane skeleton are referred to as actin filaments. Is there any evidence that these filaments are comprised solely of actin? It would seem more likely that they are spectrin cross-members linking into actin junction points, as imaged in previous cryoEM studies [1].

1. Nans, A., N. Mohandas, and D.L. Stokes, Native ultrastructure of the red cell cytoskeleton by cryo-electron tomography. *Biophys J*, 2011. 101(10): p. 2341-50.

Reply: We agree with the reviewer and have added the following sentence, including the mentioned reference, to the manuscript to clarify this issue.

"In uninfected HbAA erythrocytes, actin is organized as short protofilaments of 30-40 nm that, together with other factors, form junctional complexes which crosslink spectrin tetramers into hexagonal arrays²³." (results: page 5, second paragraph).

REVIEWERS' COMMENTS:

Reviewer #1 (Remarks to the Author):

The authors have adequately addressed my comments and incorporated my suggestions.

Reviewer #2 (Remarks to the Author):

Upon careful reviews with the reply the authors conveyed, it is determined that they have sufficiently and adequately responded to each comment.

Reviewer #3 (Remarks to the Author):

The authors have answered my queries satisfactorily

NCOMMS-16-00821A

Oxidative insult can induce malaria-protective trait of sickle and foetal erythrocytes

REVIEWERS' COMMENTS:

Reviewer #1 (Remarks to the Author):

The authors have adequately addressed my comments and incorporated my suggestions.

Reviewer #2 (Remarks to the Author):

Upon careful reviews with the reply the authors conveyed, it is determined that they have sufficiently and adequately responded to each comment.

Reviewer #3 (Remarks to the Author):

The authors have answered my queries satisfactorily

Reply: We thank the reviewers for their support and are grateful that they have accepted the manuscript in its present form.